# Enhancement of Nutritional Substance, Trace Elements, and Pigments in Waxy Maize Grains through Foliar Application of Selenite

**DOI:** 10.3390/foods13091337

**Published:** 2024-04-26

**Authors:** Boyu Lu, Haoyuan An, Xinli Song, Bosen Yang, Zhuqing Jian, Fuzhu Cui, Jianfu Xue, Zhiqiang Gao, Tianqing Du

**Affiliations:** 1College of Agriculture, Shanxi Agricultural University, Jinzhong 030801, China; 18735424089@163.com (B.L.);; 2Ministerial and Provincial Co-Innovation Centre for Endemic Crops Production with High-Quality and Efficiency in Loess Plateau, Shanxi Agricultural University, Jinzhong 030801, China

**Keywords:** waxy maize biofortification, milk stage, selenite, functional component content, nutritional quality, trace elements

## Abstract

Selenium (Se) is a micronutrient known for its essential role in human health and plant metabolism. Waxy maize (*Zea mays* L. *sinensis kulesh*)—known for its high nutritional quality and distinctive flavor—holds significant consumer appeal. Therefore, this study aims to assess the effects of foliar Se spraying on the nutritional quality of waxy maize grains, with a focus on identifying varietal differences and determining optimal Se dosage levels for maximizing nutritional benefits. We employed a two-factor split-plot design to assess the nutritional quality, trace elements, and pigment content of jinnuo20 (J20) and caitiannuo1965 (C1965) at the milk stage after being subjected to varying Se doses sprayed on five leaves. Our findings indicate superior nutrient content in J20 compared to C1965, with both varieties exhibiting optimal quality under Se3 treatment, falling within the safe range of Se-enriched agricultural products. JS3 (0.793) demonstrated the highest overall quality, followed by JS2 (0.606), JS4 (0.411), and JS1 (0.265), while CS0 had the lowest (−0.894). These results underscore the potential of foliar biofortification to enhance the functional component contents of waxy maize grains.

## 1. Introduction

Selenium (Se) is an important trace element present in animals and humans, serving as a cofactor for antioxidant enzymes, such as glutathione peroxidase. Its functions include bolstering the immune system, reducing cardiovascular disease, thyroid regulation, detoxification, and anticancer and antiviral effects [1]. Humans primarily acquire Se through their daily diet and/or nutritional supplements [2]. For adults, the European Food Safety Authority recommends a dietary reference intake of 70 μg/day, with a tolerable upper limit of 300 μg/day. According to the Chinese health industry standard (WS/T 578.3-2017) [3], the recommended dietary intake is 60–400 μg/day [4]. However, over 40 countries and ecological zones globally face severe Se deficiencies. Among these, 72% of the ecological zones of China are Se-deficient or exhibit low Se, leading to approximately 2/3 of the population of China experiencing varying degrees of Se deficiency [5]. Cereal crops serve as the primary source of dietary Se for humans. Nonetheless, their Se content is typically low, ranging from 0.01 to 0.55 μg/g in cereals [6]. Therefore, enhancing Se levels in food crops to improve human Se nutritional intake is essential and urgently required.

The agronomic biofortification of Se in food crops emerges as a promising solution to combat Se deficiency in science popularization [1]. However, its efficacy hinges on formulation, dosage, and timing [7]. Among the various methods, foliar and soil applications are the most prevalent for Se agronomic bioaugmentation [8]. However, foliar spraying proves more effective than soil application in elevating Se levels in cereal crops [9]. This is because soil application results in significant Se wastage, with 80–95% of selenate potentially being lost through irrigation or rainfall. This poses risks of groundwater pollution and causes potential environmental hazards [10]. Moreover, the Se bioavailability in soil is influenced by soil properties, including redox potential, pH, and organic matter [11]. For example, selenite tends to bind strongly to soil iron oxides/hydroxides and organic matter, leading to reduced Se bioavailability when present in large quantities in the soil [12]. Foliar spraying facilitates direct Se transportation from leaves to grains, contrasting with soil application, where Se moves from roots to aboveground parts, which increases its higher bioavailability [2]. Selenite and selenate are commonly used forms of Se application. Research suggests that the foliar application of selenite outperforms selenate in increasing wheat grain Se content, regardless of the spraying rate [4]. Moreover, recent studies indicate that spraying Se—whether selenate or selenite—on crop leaves during growth improves the biological enhancement effect of Se. This is due to the easy redistribution of Se within the crops to the grains. Furthermore, applying Se during the later growth stages proves more effective in increasing crop Se content [13]. Consequently, the agronomic biofortification of Se—especially the foliar application of selenite—has gained widespread usage in wheat [14], rice [15], peanut [16], sorghum [17], foxtail millet [18], and others in recent years.

Se treatment can influence the metabolism of carbohydrates, proteins, and various physiological and metabolic processes, leading to alterations in nutrient content [19]. This effect has been observed in foxtail millets [20], naked oats [21], rice [22], wheat [23], and various vegetables. The foliar application of Se enhanced the content of soluble sugar, amino acids, and vitamin C in tomato fruits [24]. It is believed that this application delays fruit ripening and preserves fruit quality by reducing ethylene and reactive oxygen species (ROS) [25]. Se biofortification not only enhances the nutritional quality of crops but also significantly boosts Se content in agricultural products, aiding in mitigating Se malnutrition [26,27]. Furthermore, Se exhibits bidirectional effects on micronutrients in plants [28]. Within a specific dose range, Se promotes the absorption of iron (Fe), manganese (Mn), zinc (Zn), and copper (Cu) by plants. This phenomenon has been confirmed in wheat [29,30,31], soybeans [1], cabbage [32], tomatoes [33], and lettuce [34]. However, research indicates that Se has inhibitory effects on trace elements [35,36], which may vary depending on the Se forms, doses, application methods, and crop types [37]. Moreover, anthocyanins demonstrate strong antioxidant and free radical-scavenging properties. A high intake of crops rich in carotenoids is associated with reduced cancer risk, cardiovascular disease, coronary heart disease, and eye disorders [38,39]. Xue et al. demonstrated that Se biofortification led to increased anthocyanin levels in colored grain wheat [40]. Similarly, Dong et al. identified a significant positive correlation between carotenoids and Se dosage. When Se dosage fell within the range of 0.01–0.05 g/kg, the carotenoid content in wolfberry leaves was significantly increased [41].

Waxy maize (*Zea mays* L. *sinensis kulesh*), a distinct maize variety in China, boasts a rich cultivation history globally and widespread consumption [42,43]. Unlike ordinary maize, waxy maize offers higher protein, vitamin, trace element, carotenoid, and anthocyanin contents [44]. Furthermore, waxy maize is rich in dietary fiber and phenolic substances, promoting human intestinal peristalsis, preventing chronic diseases, and exhibiting antioxidant and free-radical-scavenging characteristics [45]. Waxy maize has a distinctive taste owing to the amylopectin in its endosperm starch, aligning well with the modern dietary needs of contemporary consumers [46]. However, the widespread soil Se deficiency globally implies that naturally Se-rich products in the current market fall short of meeting the demand for Se deficiency. This contradicts the rising living standards and human expectations for crops’ nutritional value. Hence, the primary objectives at this stage involve developing Se-rich agricultural products and enhancing the functional component content of crops. We hypothesized that foliar spraying of Se can enhance the Se content of waxy maize grains, stimulate the synthesis and absorption of other trace elements, and elicit the secondary effects of increasing nutritional substance and pigment content, thereby effectively improving the quality of maize grains. Therefore, the purpose of this study is as follows: (i) to explore whether foliar spraying with varying Se doses can enhance the levels of nutritional substances, trace elements, and pigments in waxy maize; (ii) to determine the optimal Se dose for optimizing waxy maize quality during the milky stage (25 days after pollination). The findings of this study could offer a theoretical framework for regulating Se enrichment in waxy maize grains and enhancing the content of functional components through foliar Se spraying.

## 2. Materials and Methods

### 2.1. Testing Material

The tested varieties, Jinnuo 20 (J20) and Caitiannuo 1965 (C1965), were provided by the Maize Institute of the Shanxi Academy of Agricultural Sciences.

Sodium selenite (Na_2_SeO_3_) with an analytical purity of 97%, sourced from Beijing Beihua Fine Chemicals Co., Ltd., Beijing, China, served as the selected Se source.

Ethyl alcohol, anthrone, resorcinol, ammonium molybdate, and ninhydrin were procured from Beijing Solarbio Science & Technology Co., Ltd., Beijing, China. The other required drugs were procured from Guoyao Group Chemical Reagent Co., Ltd., Beijing, China.

### 2.2. Site Description and Experimental Design

The experiment was conducted at the Farming Station of Shanxi Agricultural University (113°51′ E, 38°46′ N) during the 2019/2020 season. In 2019, the experiment commenced on 22 May, with sample collection occurring during the milky stage on 20 August. Similarly, in 2020, the experiment started on 26 May, and samples were collected during the milky stage on 22 August. 

A two-factor split-plot design was employed in the experiment. Two waxy maize varieties were assigned to the main area, while the sub-area received five Se spraying doses (calculated as pure Se). This included 0 g ha^−1^ (Se0), 11.25 g ha^−1^ (Se1), 22.5 g ha^−1^ (Se2), 45 g ha^−1^ (Se3), 90 g ha^−1^ (Se4), totaling 10 treatments, repeated thrice, leading to a total of 30 experimental plots. Each plot covered an area of 18 m^2^, with row spacing set at 60 cm, plant spacing set at 30 cm, and a planting density of 54,000 plants/hm^2^. Before sowing, the soil surface of the test field underwent thorough mixing. Soil fertility was assessed after air-drying, crushing, and sieving (Table 1). The initial Se content measured 0.041 mg/kg, indicating Se-deficient soil.

Over the 2-year experiment, 600 kg/hm^2^ of Nongyou compound fertilizer—maize formula fertilizer N-P_2_O_5_-K_2_O 26-9-5, total nutrient ≥40%—served as the base fertilizer before sowing. At the trumpet period, the deep-buried fertilization method involved the application of 300 kg/hm^2^ of urea (total nitrogen (N) ≥ 46%), and irrigation followed the topdressing application. During waxy maize growth, Se was sprayed onto the leaves three times: during the jointing-trumpet, trumpet-tasseling, and early filling stages. Each spraying event delivered one-third of the predetermined total amount. A sprayer was used to apply the corresponding amount of sodium selenite (Na_2_SeO_3_) solution onto the leaf surfaces after 5 o’clock in the afternoon on sunny days. During the spraying process, a plastic film was utilized to isolate each small area, preventing accidental spraying onto adjacent regions.

### 2.3. Sample Collection and Determination

At the milky stage, ears of waxy maize were sampled, with 10 ears randomly selected from each plot for threshing and mixing. The waxy maize grains were then oven-dried at 105 °C for 30 min and further dried to a constant weight at 80 °C. Subsequently, the grains were crushed, sieved, and analyzed to determine their functional components.

### 2.4. Determination of Nutritional Quality Content

#### 2.4.1. Determination of Soluble Sugar and Sucrose Content

Waxy maize grain powder (0.1 g, dry sample) was weighed and mixed with 4 mL of 80% ethanol. The mixture was vigorously shaken and then subjected to a water bath at 80 °C for 30 min. After cooling to room temperature, the solution was centrifuged at 3000 rpm for 15 min, and the supernatant was collected. This process was conducted in triplicate and the collected supernatants were combined, filtered through activated carbon, and diluted to a volume of 50 mL. A diluted sample solution was utilized to determine soluble sugar and sucrose contents. Soluble sugar content was measured using anthrone colorimetry [47]. One milliliter of the sample solution was transferred into a glass test tube with a stopper. To this, 1 mL of distilled water and 4 mL of 0.2% anthrone solution were added. The mixture was thoroughly shaken and then heated in a boiling water bath for 15 min. Following that, the tube was removed from the water bath and allowed to cool to 25 °C. Soluble sugar content was determined colorimetrically at 620 nm using a ultraviolet spectrophotometer (Thermo Fisher Scientific Inc., Waltham, MA, USA). The sucrose content was determined using the resorcinol method [48]. First, 0.9 mL of the sample solution was transferred into a glass test tube with a stopper. Subsequently, 0.1 mL of 2 N NaOH solution was added, and the mixture was shaken before being heated in a boiling water bath for 10 min. After cooling to room temperature, 1 mL of 0.1% resorcinol solution and 3 mL of HCl solution were added. The mixture was shaken and then heated in a water bath at 80 °C for 60 min. After cooling to room temperature, the tubes were removed, and the absorbance was measured at a wavelength of 500 nm.

#### 2.4.2. Determination of Starch Content and Its Components

Starch content was determined using anthrone colorimetry [47]. Initially, the sample underwent degreasing, after which 0.1 g of a dry sample was weighed. The same extraction steps were used for soluble sugar and sucrose to obtain the supernatant. However, only precipitation data were utilized. The resulting precipitate was transferred into a 100 mL conical flask, to which 10 mL of 6 mol/L HCl was added. The mixture was then heated in a boiling water bath for 10–20 min. The degree of hydrolysis was assessed using an iodine reagent until it no longer turned blue. The conical flask was removed and allowed to cool to room temperature before adding 20 mL of distilled water. After thorough shaking, it was filtered into a 50 mL volumetric flask and diluted to a constant volume. From the volumetric flask, 1 mL of the liquid was taken from the stoppered glass test tube, and 4 mL of anhydrous ethanol was added. Following vigorous shaking, 1 mL of the liquid was removed from the stoppered glass test tube, and the anthrone method, similar to that used for determining soluble sugar content, was employed. The amylose content was assessed using the dual-wavelength method [49]. Initially, 0.1 g of the degreased sample was weighed in a beaker, followed by the addition of 10 mL of 0.5 mol/L KOH solution. The sample was then heated in a boiling water bath for 10 min. After cooling to room temperature, the sample was diluted in a 50 mL volumetric flask with distilled water and allowed to stand for 15 min before filtration. Subsequently, 5 mL of the filtrate was mixed with 25 mL of distilled water. The pH was adjusted to approximately 3.5 using 0.1 mol/L HCl solution, followed by the addition of 0.5 mL of iodine reagent. The mixture was diluted to 50 mL with distilled water and allowed to stand for 25 min. Distilled water served as a blank control for absorbance determination. Amylopectin content = starch content-amylose content.

#### 2.4.3. Determination of Protein Content

A dry sample of waxy maize grain was weighed and placed in a conical flask using the biuret method [50]. Subsequently, 0.5 mL of carbon tetrachloride, 5 mL of 0.5 mol/L KOH solution, and 20 mL of biuret reagent were added. After oscillating on the oscillator for 10 min and standing at room temperature for 30 min, an appropriate volume of solution was transferred to a centrifuge tube. The mixture was then centrifuged at 3500 rpm for 10 min, and the supernatant was collected for colorimetric analysis at 550 nm. Additionally, a blank control was prepared by mixing 2 mL of 0.05 mol/L KOH solution and 8 mL of biuret reagent.

#### 2.4.4. Determination of Fat Content 

Fat content was determined using a Soxhlet extractor according to the Chinese National Standard method (GB/T 5009.6-2016) [47,51]. A total of 3.0 g (m_2_) of maize powder was weighed with great precision and placed in the filter paper tube. The tube was then inserted into the extraction tube of the Soxhlet extractor and connected to a receiving bottle (m_0_) which had been dried to a constant weight. At the top of the condensing tube of the extractor, 2/3 of the volume of the bottle was filled with light petroleum. The light petroleum was continuously refluxed and extracted for 8 h by heating in a water bath. The receiver bottle was removed and the petroleum ether was recovered. When 1–2 mL of solvent remained in the receiving flask, it was evaporated in a water bath, dried at 105 °C for 1 h, cooled in a dryer for 0.5 h, and weighed (m_1_). The water content was calculated using the following equation:(1)$$Crude\;fat\;content\;(\%)=\frac{m_{1}-m_{0}}{m_{2}}\times 100$$

#### 2.4.5. Determination of Vitamin C Content

The molybdenum blue colorimetric method was used [47]. One gram of dry waxy maize grain sample was weighed, and oxalic acid EDTA solution was added to a constant volume in a 25 mL volumetric flask. A portion of the homogenate was centrifuged at 3000 rpm for 10 min, and 10 mL of the supernatant was taken. Subsequently, 1 mL of metaphosphoric acid–acetic acid solution, 2 mL of 5% sulfuric acid solution, and 4 mL of 5% ammonium molybdate solution were sequentially added. The volume was constant in a 25 mL volumetric flask with distilled water. After shaking, the mixture was allowed to stand for 15 min, and the color was compared at a wavelength of 705 nm.

#### 2.4.6. Determination of Lysine Content

Ninhydrin colorimetric method was used [50]. Briefly, 0.1 g of defatted waxy maize grain dry sample was shaken in 100 mL of distilled water at room temperature for 20 min. The mixture was filtered or centrifuged, and 0.5 mL of the resulting supernatant was transferred to a stoppered glass test tube. A blank control was prepared with 0.5 mL of distilled water. Subsequently, 1.5 mL of ninhydrin solution was added, and the mixture was heated in a boiling water bath for 20 min. After cooling to room temperature, 8 mL of 80% absolute ethanol was added, followed by thorough shaking, and compared at a wavelength of 530 nm.

### 2.5. Determination of Trace Elements

Waxy maize grain samples (0.2 g) were weighed. They were then digested using an LWY84B alimentary furnace reaction system, (Wuhan Gremo testing equipment Co., Ltd., Wuhan, China) with a nitric acid to hydrogen peroxide ratio of 6:2. After cooling, the sample was diluted to 10 mL and filtered. The levels of Se, Fe, Mn, Cu, and Zn were analyzed using Agilent 7700x inductively coupled plasma mass spectrometry (Agilent Technology Co., Ltd., Lexington, KY, USA) [40].

### 2.6. Determination of Pigment Content

#### 2.6.1. Determination of Anthocyanin Content

The anthocyanin content was assessed using the pH differential method [52]. A specific amount of waxy maize grain powder (dry sample) was weighed. An acidic ethanol solution (pH = 2.5) was added at a solid–liquid ratio of 1:15. The solution underwent ultrasonic extraction for 30 min, followed by centrifugation, and the supernatant was collected. This process was repeated twice, and the supernatants were combined and diluted in a 25 mL volumetric flask. From the resulting solution, two aliquots of 2 mL liquid were taken from the supernatant and diluted to 10 mL with buffers of pH 1 and 4.5, respectively. These samples were then reacted in a 40 °C water bath for 30 min. Distilled water served as the blank control and the colors were measured at 519 and 700 nm.

#### 2.6.2. Determination of Carotenoid Content

Determination of carotenoid content: The acetone petroleum ether extraction method was used [53]. A total of 0.2 g of waxy maize grain powder (dry sample) was weighed, and 4 mL of extract (acetone: petroleum ether = 4:1) was added. The solution was then shaken and subjected to ultrasonic extraction for 30 min in the dark, maintaining a temperature of 50 °C. After ultrasonic treatment, the solution was centrifuged at 10,000 rpm for 10 min, and the supernatant was collected. The process was repeated to extract the residue again. The supernatants were combined and diluted to 10 mL, and the color was compared at the maximum absorption peak wavelength.

### 2.7. Statistical Analysis

All data are presented as the mean ± standard error. Statistical analysis was performed using SPSS21.0 Statistics software. Origin 2021 software was used for mapping. One-way analysis of variance with Duncan’s multiple range test was to determine significant differences among the treatments, with a statistical significance level at *p* ≤ 0.05. Correlation analysis was performed using the Pearson method, and the results with corresponding probability values of *p* < 0.05, *p* < 0.01, and *p* < 0.001 indicated different degrees of correlation. For principal component analysis (PCA), the maize functional nutrient quality was normalized to create a correlation matrix. Eigenvalues and relative contribution rates were determined from the correlation matrix, and the factor scores of the principal components were calculated [54].

## 3. Results

### 3.1. Nutritional Quality

Significant differences in the nutritional quality of waxy maize were observed across different Se foliar spraying treatments (Figure 1A–P).

The foliar application of Se led to an increase in soluble sugar content in both varieties of waxy maize, with C1965 exhibiting higher levels than J20. In 2019, the highest soluble sugar content of J20 and C1965 was recorded under Se3 treatment, reaching 75.56 mg/g and 143.79 mg/g, respectively. Similarly, in 2020, J20 and C1965 showed maximum values under the Se4 treatment, marking a significant increase of 20.17% and 20.31%, respectively, compared to the control. However, no significant differences were observed between the Se3 and Se4 treatments (Figure 1A,B).

Figure 1C,D show that the sucrose content of J20 gradually increased with increasing Se spraying dose, peaking under the Se4 treatment. In 2019 and 2020, increments of 35.91% and 23.03%, respectively, were exhibited compared to the control. Conversely, for C1965, sucrose content initially increased and then declined with an increase in the Se spraying dose, reaching its peak under the Se3 treatment. In 2019 and 2020, it increased by 24.34% and 18.03%, respectively, compared to the control, with no significant difference observed compared to Se4.

As the Se spray dose increased, the vitamin C content of J20 gradually rose. Under Se4 treatment, the vitamin C content in J20 peaked, exhibiting significant increases of 21.16% (2019) and 26.23% (2020), respectively, compared to Se0. In 2019, the vitamin C content of C1965 initially increased and then decreased with an increasing Se spraying dose, reaching its highest level at Se2, which saw a significant increase of 21.94% compared to the control. However, in 2020, the vitamin C content of C1965 under Se4 treatment significantly surpassed that of other treatments, showing increases of 32.56%, 22.72%, 23.42%, and 14.65% compared to Se0, Se1, Se2, and Se3, respectively (Figure 1E,F).

The protein and lysine content of waxy maize grains displayed an initial increase followed by a decrease with an increasing Se spraying dose. For J20, the protein and lysine contents were the highest under Se3 treatment. This led to a significant increase in protein by 26.15% (2019) and 21.30% (2020) and lysine by 15.43% (2019) and 13.44% (2020), respectively, compared to Se0. Compared to the control, foliar spraying of Se significantly increased the protein content of C1965 in 2019 by 32.5%, 32.41%, 31.07%, and 22.98%, respectively. This is compared to the control, with no significant difference observed between treatments. In 2020, the protein content of C1965 peaked under the Se3 treatment, showing a significant increase of 25.46% compared to the control, with no significant difference among the Se0, Se1, Se2, and Se4 treatments. However, no significant differences were observed in the lysine content of C1965 among the treatments (Figure 1G–J).

After spraying selenite onto the leaves, the fat and protein contents of the maize grains exhibited a similar pattern. In 2019, the fat contents of J20 and C1965 treated with Se2 reached their peaks, significantly surpassing those of Se0 by 4.94% and 9.34%, respectively. In 2020, the fat content of maize grains treated with Se3 peaked, showing significant increases of 12.10% (J20) and 10.74% (C1965) compared to Se0 (Figure 1K,L).

As the selenite-spraying dose increased, the total starch, amylose, and amylopectin contents in maize initially rose and then declined. Under the Se3 treatment, J20 exhibited the highest total starch, amylose, and amylopectin contents, with significant increases of 12.30%, 18.36%, and 12.04% (2019), 12.23%, 24.90%, and 11.87% (2020) compared to Se0. However, no significant differences were observed in the total starch, amylose, and amylopectin contents of C1965 in 2019. In 2020, C1965 displayed a similar trend of initially increasing and then decreasing total starch, amylose, and amylopectin contents with the increasing Se dose. This reached maximum levels under the Se3 treatment, which led to a significant increase of 7.62%, 18.43%, and 7.19%, respectively, with no significant difference observed between the Se0 and Se1 treatments (Figure 1M–P).

### 3.2. Trace Elements

The foliar spraying of selenite elevated the Fe content of waxy maize grains. In 2019, the J20 and C1965 initially exhibited an increase followed by a decline in Fe content with increasing foliar spraying Se dose. The peak for both varieties was observed under the Se2 treatment, showing a significant increase of 7.83% and 8.31% compared to the control, respectively. In 2020, J20 exhibited a significant 8.57% increase in Fe content under Se2 treatment compared to the control, with no significant difference among Se1, Se2, Se3, and Se4. Similarly, the Fe content of C1965 followed a similar pattern of an initial increase followed by a decrease with an increase in Se dose. This peaked at Se2, with a significant 8.61% increase compared to the control. No significant differences were observed between Se0, Se1, Se3, and Se4 treatments (Figure 2A,B).

With increasing Se spray dose, the Mn content of waxy maize grains exhibited an initial rise followed by a decline. In 2019, Se2 treatment yielded the highest Mn content for J20 and C1965, at 6.95 μg/g and 6.04 μg/g, respectively. However, in 2020, the peak Mn content for J20 and C1965 occurred under Se1 treatment, showing significant increases of 13.91% and 9.57%, respectively, compared to the control. Conversely, under Se4 treatment, Mn content significantly decreased by 13.28% for J20 and 5.73% for C1965 compared to the control (Figure 2C,D).

As the Se spraying dose increased, the Cu contents of J20 and C1965 initially rose before declining, peaking at Se2 treatment. Compared to the control, J20 exhibited significant increases of 33.31% and 12.52% in Cu content in 2019 and 2020, respectively. However, C1965 showed increases of 27.11% and 10.88% in 2019 and 2020. In 2020, the Cu content of J20 and C1965 under Se4 treatment was lower than that of the control, with decreases of 1.17% and 7.72%, respectively (Figure 2E,F).

As the Se spraying dose increased, the Zn content of J20 and C1965 initially rose before declining, reaching its peak under Se3 treatment. Compared to the control, J20 exhibited increases of 13.60% and 7.10% in 2019 and 2020, respectively. However, C1965 showed increases of 14.95% and 11.33% in 2019 and 2020, respectively (Figure 2G,H).

As the Se spraying dose increased, the Se content of waxy maize grains gradually rose, reaching its peak under the Se4 treatment. In 2019 and 2020, compared to the control, the Se content of J20 significantly increased by 7.08 and 89.78 times under Se4 treatment. This represents increases of 708.63% and 897.78%, respectively. Similarly, the Se content of C1965 significantly increased by 6.13 and 10.16 times under Se4 treatment. To establish the safe Se application threshold for J20 and C1965 at the milk stage in 2019 and 2020, the dose of Se sprayed on the leaves was utilized as the independent variable X, and the average Se content in the grains served as the dependent variable Y. The data in Figure 2I,J were fitted to derive the following equations:

2019: (2)$$Y_{J20}=0.01613x^{2}+3.16122x+57.80307\;R^{2}=0.9991$$
(3)$$Y_{C1965}=0.00889x^{2}+4.11702x+71.74792\;R^{2}=0.9971$$

2020: (4)$$Y_{J20}=0.0214x^{2}+1.0878x+30.36923\;R^{2}=0.9991$$
(5)$$Y_{C1965}=0.0029x^{2}+3.91081x+43.44097\;R^{2}=0.9874$$

According to the supply and marketing cooperation industry standard of the People’s Republic of China’s Se-rich agricultural products (GHT1135-2017) [55], the acceptable range for total Se content in cereals is 0.10–0.50 mg/kg. If fresh food is consumed during the milk stage, the estimated Se spraying doses on the leaves of J20 in 2019 and 2020 range from 12.61 to 94.41 g/ha and 37.12 to 124.89 g/ha, respectively. For C1965 during the same periods, the foliar spraying Se doses of C1965 were calculated to be 6.85–87.49 g/ha, and 14.32–108.08 g/ha, respectively. The Se content in waxy maize grains from this experiment did not exceed the safety upper limit of 500 μg/kg.

### 3.3. Pigments

The anthocyanin content was observed to be higher in J20 than that in C1965. The anthocyanin content in waxy maize grains exhibited an upward trend with increasing foliar spraying Se dose, peaking under the Se4 treatment (Figure 3A,B). In 2019 and 2020, J20 exhibited a significant increase in anthocyanin, of 16.96% and 14.60%, respectively, under Se4 treatment. However, no significant difference was observed between Se0, Se1, Se2, and Se3. Similarly, in 2019 and 2020, C1965 experienced a significant increase in anthocyanin, of 37.68% and 27.64%, respectively, under Se4 treatment, with no significant difference between Se2, Se3, and Se4.

As the foliar spraying Se dosage increased, the carotenoid content in waxy maize grains exhibited a pattern of initially increasing and then decreasing. All reached their peak at Se3. The carotenoid content of C1965 surpassed that of J20. Under the Se4 treatment, the carotenoid content of J20 experienced a significant increase of 36.18% and 45.08% in 2019 and 2020, respectively, compared to the control. Similarly, the carotenoid content of C1965 saw a significant increase of 21.74% and 63.17% (Figure 3C,D).

### 3.4. Correlation Analysis

Correlation analysis of the 16 nutritional quality indicators revealed significant relationships among these parameters (Figure 4). Vitamin C exhibited a negative correlation with soluble sugar, sucrose (*p* < 0.01), and carotenoids (*p* < 0.05), while showing positive correlations with protein, Fe (*p* < 0.01), anthocyanin, lysine, and Zn (*p* < 0.001). Anthocyanin demonstrated positive correlations with protein (*p* < 0.05), lysine (*p* < 0.01), Fe, and Zn (*p* < 0.001), and negative correlations with soluble sugar, sucrose, and carotenoids (*p* < 0.001). Furthermore, Fe and Zn exhibited significant correlations with multiple nutritional indicators in maize grains. Conversely, Se content demonstrated significant correlations only with sucrose (0.45) and Mn (−0.46).

### 3.5. Principal Component Analysis

Principal component analysis was conducted on the quality indices of the two maize varieties with five Se doses in 2019 and 2020. Four principal components with eigenvalues > one were extracted. In 2019, the cumulative contribution rate of the first two principal components was 63.792%, while in 2020, it was 67.872%, indicating high representativeness (Appendix A). The first two principal components (PC1 and PC2) were utilized to create a score-load diagram. Different maize varieties were distinguished on either side of the PC1 axis, with J20 grains exhibiting higher contents of starch, vitamin C, anthocyanin, and other nutrients. The grains of C1965 exhibited higher contents of Se, carotenoids, and soluble sugars. Furthermore, the distinction between low- and high-dose Se treatments was evident on both sides of the PC2 axis. While the high-dose Se treatment shows more abundant nutrients, the low-dose treatment lacks nutrients in comparison (Figure 5). A comprehensive scoring of maize nutritional quality under different treatments over the 2 years was comprehensively assessed. It was observed that the nutritional quality of J20 surpassed that of C1965, with both varieties demonstrating the best quality characteristics under the Se3 treatment (Appendix A). JS3 (0.793) displayed the highest comprehensive quality, followed by JS2 (0.606), JS4 (0.411), and JS1 (0.265). CS0 exhibited the lowest comprehensive quality (−0.894).

## 4. Discussion

Studies have shown that the foliar spraying of Se can promote the content of nutritional quality of crop grains, and our experiments obtained similar results [21,22,23,24,25]. This effect may stem from Se indirectly stimulating an increase in acid invertase activity, boosting chlorophyll content, and enhancing leaf photosynthetic rate. This results in an increase in sugar accumulation at the same time [56]. Simultaneously, Se plays a role in the carbohydrates, amino acids, and secondary metabolism of proteins, influencing the activity of biological enzymes in metabolic processes. Consequently, this leads to increased soluble sugar and sucrose content in waxy corn grains, thereby enhancing crop nutritional quality [20]. Se can increase selenoprotein levels in plants while mitigating vitamin C oxidation by hydrogen peroxide and lipid peroxide in plant cells, thus boosting vitamin C content [57,58]. The research indicates that an appropriate Se dose can significantly enhance crude protein, lysine, fat, and starch levels in foxtail millet, with the optimal results observed during the filling stage [59]. Moreover, Se can directly engage in protein synthesis by serving as an Se-substituted amino acid precursor and can function as a component of tRNA to facilitate amino acid transport for protein synthesis. Moreover, high-Se treatment disrupts the normal physiological metabolism of waxy maize grains, leading to a decline in protein content [60]. In our study, we observed that crude protein content initially increased and then decreased with foliar Se application. This pattern may be attributed to the gradual incorporation of selenoamino acids into selenoproteins synthesis when Se levels in the plant are low. However, excessive Se doses in the body could exert mild toxic effects and reduce the Se protein content in the grains. This observation suggests that the maximum Se dosage in this experiment reached a significantly high level. Prior research indicates that foliar Se application can significantly enhance lysine and crude starch levels in crops while exerting no significant effect on crude fat content. Our study revealed an increase in fat content following Se foliar spraying. However, the observed rise in lysine content in C1965 and starch in 2019 was not statistically significant, and was possibly influenced by maize varieties, soil environment, and Se fertilizer dosage.

The application of Se not only enhances the nutritional value uptake by crops but also influences the absorption of Se and other mineral elements to varying extents. Consequently, Se-enriched maize could serve as a dietary Se source, potentially mitigating the health issues associated with Se deficiency. Our findings revealed that C1965 exhibited a stronger Se enrichment capability than that of J20. However, J20 outperformed C1965 in terms of comprehensive indicators. Nonetheless, an excessive accumulation of Se can pose toxicity risks to humans and plants. In higher plants, low Se doses can enhance crop growth and stress resistance. However, excessive Se levels can harm crops and even lead to a significant decrease in yield [29,61]. However, such detrimental effects were not observed in our study. Even with the highest Se foliar spray dosage employed—where C1965 exceeded the Se-enriched maize standard of 0.0143 mg/kg in 2019—no damage, such as leaf-surface yellowing, occurred. This absence of adverse effects may be attributed to variations in Se fertilizers, application doses, and crop tolerance to Se. Further analysis of the equation suggests that within the soil Se content background range of 0.041–0.057 mg/kg, consuming fresh food during the milky stage poses no risks. The Se spraying doses for J20 and C1965 ranged from 37.12 to 94.41 g/hm^2^ and 14.32 to 87.49 g/hm^2^, respectively. Low Se doses promote plant growth and enhance the uptake of trace elements such as Fe, Mn, Cu, and Zn. Conversely, higher doses can be toxic to plants and impede increases in trace elements [28,62,63]. This experiment adhered to these principles. Data from both years indicated that maize grain exhibited the highest Fe content, followed by Zn, while Mn and Cu contents were comparatively lower. However, the maximum Se spraying dose for each element varied. Different doses of SeO_3_^2−^ exert distinct effects on the absorption of cations such as Fe^2+^, Mn^2+^, Cu^2+^ and Zn^2+^ [29]. The effect of foliar Se spraying on mineral elements in grains can be influenced by several factors, including ion synergy and antagonistic effects. Other factors include the involvement of various enzymes associated with protein synthesis and the physiological traits of the crops themselves. The interactions between Se and micronutrients in diverse plant species remain controversial and require further study [29,63,64].

Anthocyanins and carotenoids are pigments known for the characteristic blue-purple and yellow-orange hues of their grains, respectively. However, their presence in crop grains is limited [65]. Se treatment has been reported to increase the levels of anthocyanin and carotenoid [40,58]. In line with previous findings, the results of this study showed that J20 exhibited a higher anthocyanin content than that of C1965. Furthermore, the anthocyanin levels in waxy maize grains increased with higher foliar Se doses. This phenomenon may stem from alterations in the expression levels of genes involved in anthocyanin synthesis, including *PAL*, *4CL*, *CHS*, *F3H*, *DFR*, *ANR*, and *R2R3-MYB*, induced by Se treatment. These changes could lead to the accumulation of anthocyanin metabolites such as Cy-3-O-(6-O-malonyl)-glu and Pn-3-O-(6-O-malonyl)-glu in grains. Consequently, it accelerates anthocyanin synthesis while slowing down a pathway branch associated with lignin and proanthocyanidin synthesis [40]. Islam et al. discovered that Se biofortification increased carotenoid content [58]. This experiment yielded similar results, with waxy maize grains reaching their peak at Se3. C1965 exhibited significantly higher carotenoid levels than J20. The application of Se enhances the biosynthesis of photosynthetic pigments in plants by repairing chloroplasts damaged by environmental stress and ROS [66,67,68]. Studies have indicated that the potential rise in respiration rate attributed to Se application might contribute to the increase in the biosynthesis of photosynthetic pigment [69]. Furthermore, *PSY* is one of the enzymes involved in the carotenoid biosynthetic pathway. Se aids in the downregulation of *CHALSE* and the upregulation of *PSY* and *CHS*, thereby safeguarding plant pigments [53].

Correlation analysis offers straightforward operations and provides clear insights into the relationships among traits, aiding in assessing the correlation among various attributes of experimental materials. Therefore, we performed a correlation analysis of 16 nutritional quality indicators of maize under varying Se doses. Our analysis revealed significant correlations among nutritional components, trace elements, and pigments. Soluble sugar and sucrose levels exhibited significant negative correlations with vitamin C, anthocyanins, Fe, Zn, and other indicators. The observation indicates that sugar accumulation may hinder the accumulation of trace elements and functional components, aligning with the findings by Dragcevic et al. [70]. Moreover, vitamin C, protein, lysine, and fat showed positive correlations with pigments, trace elements, and nutrients, suggesting their potential contribution to quality improvements. While many prior studies have analyzed quality indicators individually, this approach may not provide a comprehensive evaluation of waxy maize quality. The principal component analysis serves as an objective, simple, and rapid evaluation system, effectively reflecting the comprehensive quality of waxy maize following Se foliar spraying [71]. Similarly, Shi et al. employed this method using seven varieties and twelve kinds of nitrogen-treated rice as research materials to evaluate seventeen quality traits, ultimately establishing a comprehensive evaluation equation for rice quality in Hubei Province, China [72]. This study revealed that J20 exhibited higher levels of starch, vitamin C, anthocyanin, and other nutrients, while C1965 displayed higher contents of Se, carotenoids, and soluble sugar. The high-dose Se treatment led to more abundant nutrients, whereas the low-dose treatment yielded relatively few nutrients. Overall, both varieties showed better comprehensive evaluation scores of waxy maize qualities under the Se3 treatment.

## 5. Conclusions

Our study demonstrated that the foliar spraying of Se increased the levels of nutritional substances, trace elements, and pigments in waxy maize grains during the milky stage. J20 exhibited higher concentrations of starch, vitamin C, anthocyanin, and other nutrients, as well as higher levels of Se, carotenoids, and soluble sugars compared to C1965 grains. In addition, the high-dose Se treatment resulted in more abundant nutrients, whereas the low-concentration treatment yielded relatively few nutrients. PCA revealed that the nutritional quality of J20 surpassed that of C1965. Both varieties exhibited optimal quality characteristics under the Se3 treatment, remaining within the safe range for Se-enriched agricultural products. 

## Figures and Tables

**Figure 1 foods-13-01337-f001:**
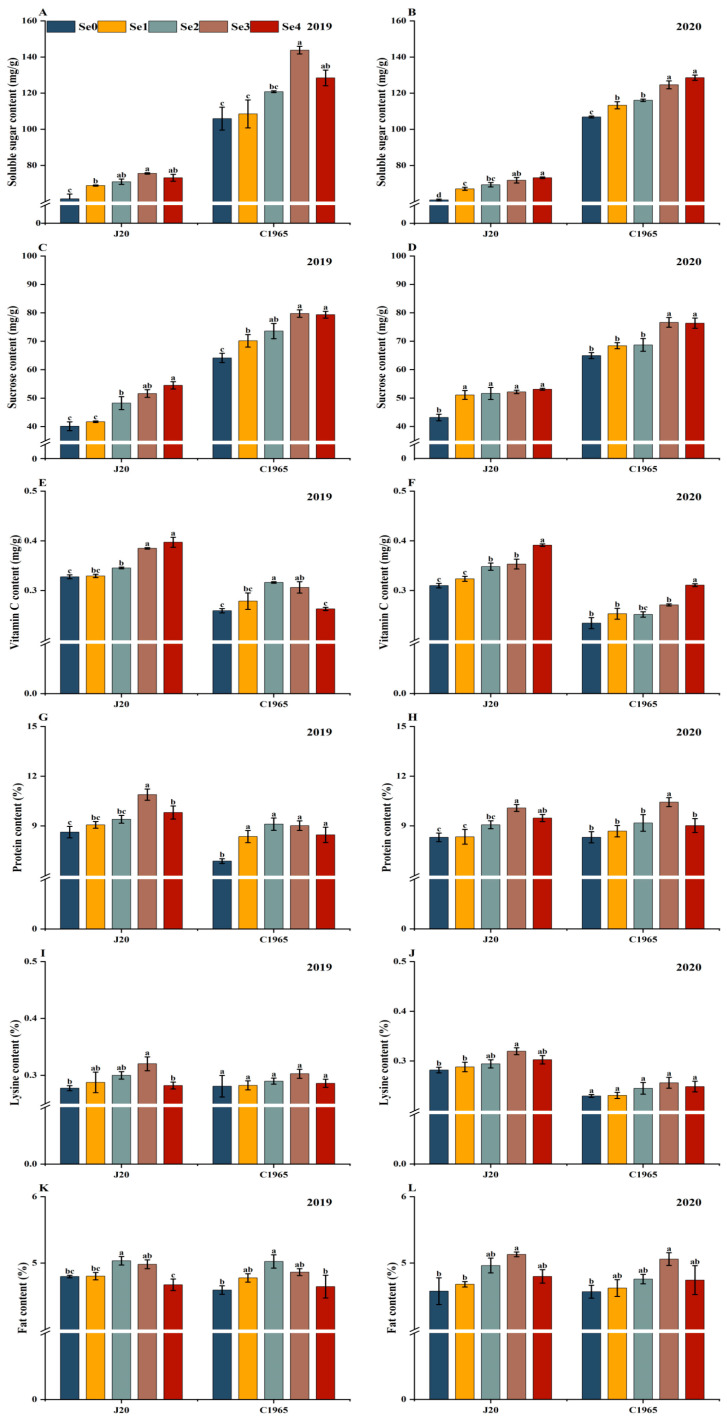
Nutritional quality analysis of maize grains following foliar application of selenite at the milky stage. Nutritional parameters examined include soluble sugar (**A**,**B**), sucrose (**C**,**D**), vitamin C (**E**,**F**), protein (**G**,**H**), lysine (**I**,**J**), fat (**K**,**L**), total starch, amylose, and amylopectin (**M**–**P**). Different doses of selenite (Se0: 0 g/ha; Se1: 11.25 g/ha; Se2: 22.5 g/ha; Se3: 45 g/ha; Se4: 90 g/ha) were applied. Data are presented as means ± standard deviations. For the different Se doses of the same variety, values not displaying the same letter are significantly different (*p* < 0.05).

**Figure 2 foods-13-01337-f002:**
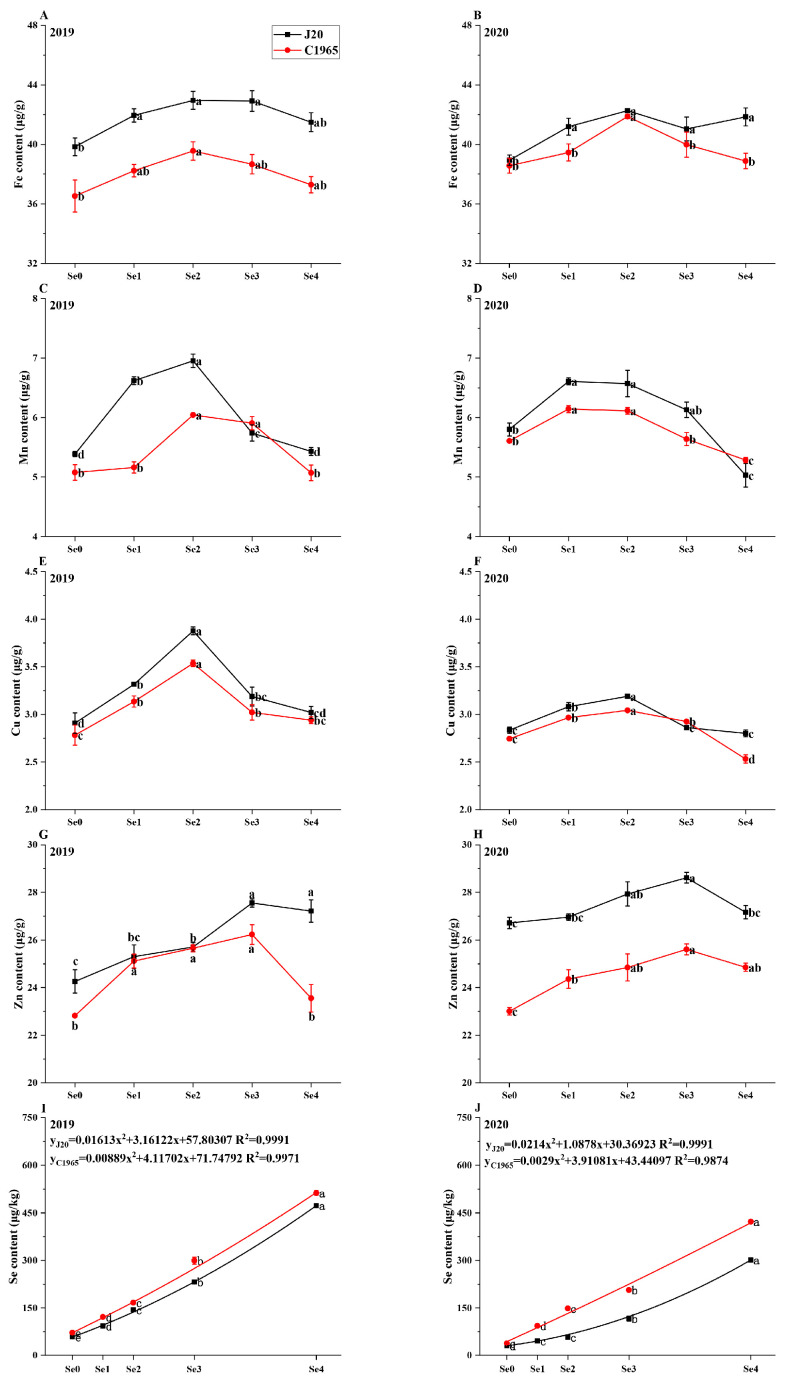
Trace element analysis in maize grains following the foliar application of selenite at milky stage. Trace elements examined include Fe (**A**,**B**), Mn (**C**,**D**), Cu (**E**,**F**), Zn (**G**,**H**), and Se (**I**,**J**). Different doses of selenite (Se0: 0 g/ha; Se1: 11.25 g/ha; Se2: 22.5 g/ha; Se3: 45 g/ha; Se4: 90 g/ha) were applied. Data are presented as means ± standard deviations. Significant differences among Se doses with the same variety are indicated by different letters (*p* < 0.05).

**Figure 3 foods-13-01337-f003:**
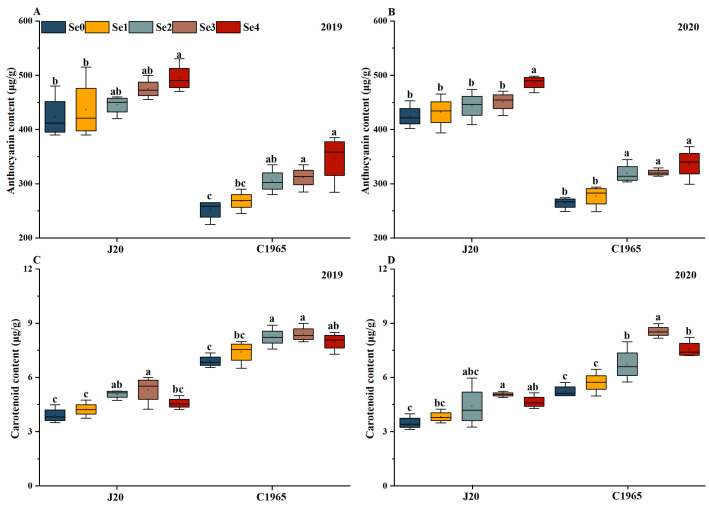
Pigment analysis in maize grains following foliar application of selenite at the milky stage. Pigments assessed include anthocyanin (**A**,**B**) and carotenoid (**C**,**D**). Different doses of selenite (Se0: 0 g/ha, Se1: 11.25 g/ha; Se2: 22.5 g/ha; Se3: 45 g/ha; Se4: 90 g/ha) were applied. Data are presented as means ± standard deviations. Significant differences among Se doses within the same variety are indicated by different letters (*p* < 0.05).

**Figure 4 foods-13-01337-f004:**
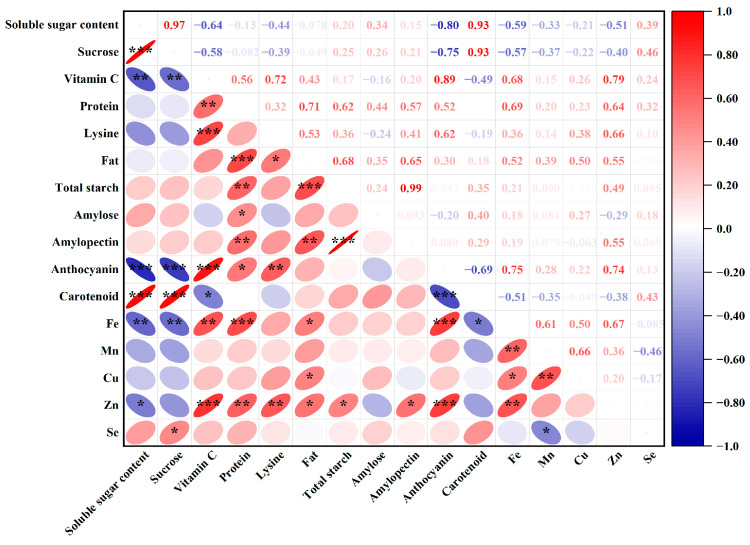
Correlation analysis of 16 nutritional qualities. *, **, and *** denote significance levels at 0.05, 0.01, and 0.001, respectively.

**Figure 5 foods-13-01337-f005:**
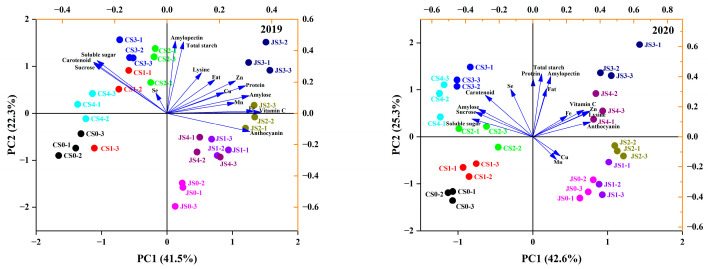
PCA of quality indicators for two maize varieties at five Se doses in 2019 and 2020. JS0 (Se:0 g/ha), JS1 (Se: 11.25 g/ha), JS2 (Se: 22.5 g/ha), JS3 (Se: 45 g/ha), JS4 (Se: 90 g/ha), CS0 (Se: 0 g/ha), CS1 (Se: 11.25 g/ha), CS2 (Se: 22.5 g/ha), CS3 (Se: 45 g/ha), CS4 (Se: 90 g/ha). −1, −2, −3 indicate repetition.

**Table 1 foods-13-01337-t001:** Soil fertility characteristics assessed at the experimental site.

Year	Organic Matter (g/kg)	Available N (mg/kg)	Available Phosphorus (mg/kg)	Available Potassium (mg/kg)	Fe (mg/kg)	Mn (mg/kg)	Cu (mg/kg)	Zn (mg/kg)	Se (mg/kg)
2019	10.8	61.96	22.43	121.05	82.13	63.86	38.13	57.37	0.057
2020	9.6	62.47	21.97	122.41	87.71	65.65	37.54	50.51	0.041

## Data Availability

The data presented in this study are available on request from the corresponding author.

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
