# Peer review of "Enhancement of Nutritional Substance, Trace Elements, and Pigments in Waxy Maize Grains through Foliar Application of Selenite"

_foods, 2024, doi:10.3390/foods13091337_

Round 1
Reviewer 1 Report
Comments and Suggestions for Authors
The manuscript "Enhancement of nutritional substance, trace elements, and pigments in waxy maize grains through foliar application of selenite"reflects an extensive and detailed work to obtain information on the effects of foliar application with Selenium in waxy corn.
There is only one aspect that deserves to be clarified and it has to do with the experimental design, the authors in line 126 say that the applications were repeated thrice, but it is not clear if it was 3 different plots randomly assigned in which the treatment was repeated. It is also not explicit whether these samples then constituted each of the triplicates or whether they were pooled to make a single sample.
Minors:
It is recommended that waxy corn be defined the first time it is mentioned in the introduction.
It is not possible to know from Table 1 whether the differences are significant.
Author Response
the responses to your comments are in the attachment.

Reviewer 2 Report
Comments and Suggestions for Authors
The paper “Enhancement of nutritional substance, trace elements, and pigments in waxy maize grains through foliar application of selenite” reports on the main findings of a two-year study on the application of Se to two different varieties of waxy maize, i.e., J20 and C1965., by foliar spraying The effect in terms of nutritional quality (Soluble Sugar and Sucrose Content, Starch Content, protein, fat, lysine content) is studied. The effect on the functional quality of waxy maize grains is also investigated (anthocyanins and carotenoids).
Before publication, I suggest few amendments. Please, find below some comments.
Introduction
Line 144: I suggest moving the clarification “(25 days after pollination)” for milky stage to line 105 where it is first mentioned.
Table 1 “available phosphorus”: “available” in capital letter
Materials and Methods
I suggest a brief description of the analytical procedure for “2.4.4. Determination of Fat Content”
Line 233: I suggest removing the hyperlink.
Section 2.6: Since in this section anthocyanins and carotenoids determination is described, I suggest splitting the paragraph into two and having 2.6.1 and 2.6.2
Results
Lines 308-309: did you make any hypothesis on that finding?
Figure 1: Unfortunately, this figure is not completely readable. The statistical significance, in particular, is too small.
Lines 362-368: I suggest amending the layout of the equations according to the Journal guidelines.
Discussion
Lines 443-445: I suggest amending the sentence so that there is a reference to the fact that this result is not applicable to ALL conditions.
Comments on the Quality of English Language
English is fine or just a minor editing now and then.
Author Response
Dear Editors and Reviewers:
Thank you very sincerely for reviewing our manuscripts and for your valuable suggestions and comments! Your suggestions will greatly improve the quality and readability of our article. After receiving your review comments, we reviewed the relevant papers and conducted serious discussions. In the spirit of exploring scientific issues and improving together, the comments are revised and answered as follows, please review. If there is anything wrong, please criticize and correct.
Thank you and best regards.
Yours sincerely,
Tianqing Du
Reviewer 2
The paper “Enhancement of nutritional substance, trace elements, and pigments in waxy maize grains through foliar application of selenite” reports on the main findings of a two-year study on the application of Se to two different varieties of waxy maize, i.e., J20 and C1965., by foliar spraying The effect in terms of nutritional quality (Soluble Sugar and Sucrose Content, Starch Content, protein, fat, lysine content) is studied. The effect on the functional quality of waxy maize grains is also investigated (anthocyanins and carotenoids).
Before publication, I suggest few amendments. Please, find below some comments.
Introduction
- Line 144: I suggest moving the clarification “(25 days after pollination)” for milky stage to line 105 where it is first mentioned.
Response: Thanks to reviewer’s suggestion, I have made the modification according to the reviewer's suggestion (lines 106-107).
- Table 1 “available phosphorus”: “available” in capital letter
Response: Thanks to reviewer’s suggestion, I have made the modification according to the reviewer's suggestion (line 145).
Materials and Methods
- I suggest a brief description of the analytical procedure for “2.4.4. Determination of Fat Content”
Response: Thanks to reviewer’s suggestion, I have modified the '2.4.4. Determination of Fat Content' (lines 210-222).
- Line 233: I suggest removing the hyperlink.
Response: Thanks to reviewer’s suggestion, I have deleted the hyperlinks (lines 247-248).
- Section 2.6: Since in this section anthocyanins and carotenoids determination is described, I suggest splitting the paragraph into two and having 2.6.1 and 2.6.2
Response: Thanks to reviewer’s suggestion, I have divided the determination of anthocyanins and carotenoids into two paragraphs, 2.6.1 and 2.6.2, respectively (lines 250 and 260).
Results
- Lines 308-309: did you make any hypothesis on that finding?
Response: Thanks reviewer’s suggestion, we made hypothesis about this finding in lines 100-103. We hypothesized that foliar spraying of Se can enhance the Se content of waxy maize grains, stimulate the synthesis and absorption of other trace elements, and elicit secondary effects of increasing nutritional substance and pigment content, thereby effectively improving the quality of maize grains.
- Figure 1: Unfortunately, this figure is not completely readable. The statistical significance, in particular, is too small.
Response: Thanks reviewer’s suggestion, I modified Figure 1 by adjusting the layout and adding breakpoints, making the statistical significance and readability higher (lines 332-334).
- Lines 362-368: I suggest amending the layout of the equations according to the Journal guidelines.
Response: Thanks to reviewer’s suggestion, I have modified the layout of the equation according to the journal guide (lines 378-387).
Discussion
- Lines 443-445: I suggest amending the sentence so that there is a reference to the fact that this result is not applicable to ALL conditions.
Response: Thanks to reviewer’s suggestion, I have modified it according to the reviewer 's opinion (lines 461-462).